# Suicide Trends in the Italian State Police during the SARS-CoV-2 Pandemic: A Comparison with the Pre-Pandemic Period

**DOI:** 10.3390/ijerph19105904

**Published:** 2022-05-12

**Authors:** Silvana Maselli, Antonio del Casale, Elena Paoli, Maurizio Pompili, Sergio Garbarino

**Affiliations:** 1State Police Health Service Department, Ministry of the Interior, 00185 Rome, Italy; silvana.maselli@poliziadistato.it (S.M.); sgarbarino.neuro@gmail.com (S.G.); 2Department of Dynamic and Clinical Psychology, and Health Studies, Faculty of Medicine and Psychology, Sapienza University of Rome, 00185 Rome, Italy; antonio.delcasale@uniroma1.it; 3Department of Neuroscience, Mental Health, and Sensory Organs (NESMOS), Faculty of Medicine and Psychology, Sapienza University of Rome, 00189 Rome, Italy; maurizio.pompili@uniroma1.it; 4Department of Neuroscience, Rehabilitation, Ophthalmology, Genetics and Maternal/Child Sciences (DINOGMI), University of Genoa, 16132 Genoa, Italy

**Keywords:** suicide, police officers, suicide rates, SARS-CoV-2 pandemic, prevention

## Abstract

The pandemic is posing an occupational stressor for law enforcement personnel. Therefore, a high priority is the need to quantify this phenomenon and put supportive programs in place. During the pandemic period, the Italian State Police implemented different support programs for the personnel. These included a national toll-free number to provide information on COVID-19 to police staff, availability of a health care service by doctors and nurses at the national level, vaccination services, working remotely, and a psychological intervention protocol called “Together we can” (“Insieme Possiamo”). Our study firstly aims to perform a descriptive analysis of the suicide in the Italian police from 2016 to 2021, and secondly aims to compare the pandemic and pre-pandemic periods. During the SARS-CoV-2 pandemic (February 2020 to October 2021), the suicide rate in the State Police did not significantly increase compared to the pre-pandemic period, showing a stable trend with a not significant decrease in the suicide rate. The implementation of staff support services by the Central Directorate of Health of the Italian State Police and individual resilience aspects of the Police personnel in response to the pandemic may have positively affected the phenomenon. These aspects pave the way to further studies on the issue to improve preventive strategies.

## 1. Introduction

Suicide is the second leading cause of death globally in the 15–29 age group [1] and is more common in men than in women [2]. The World Health Organization (WHO) estimates approximately 880,000 suicides worldwide each year, equivalent to one suicide every 40 s and one suicide attempt every 3 s. The data give the measure of a significant public health problem.

The performance of such an extreme gesture origin in a diversity of elements, as it is a multifaceted and multidimensional phenomenon. The interaction of numerous biological, genetic, psychological, social, cultural, and environmental factors has been shown in the literature [3,4,5]. Suicidal risk is often associated with psychiatric conditions related to stress, anxiety, mood disorders, family history of depression/suicide, and alcohol and substance abuse use identified as general risk factors [4,6,7]. Other possible risk factors have been associated with psychosocial variables such as unexpected and challenging life events related to health statuses, such as chronic pain [8] or severe illness [9,10,11], moments of personal and family crisis (separations, divorces) [12], job loss [13,14], economic/financial difficulties, and social isolation [15,16]. 

A study conducted by the Centers for Disease Control and Prevention in the United States (CDCs) showed a 30% increase in suicide cases in all age groups between 1999 and 2016. About 54% of cases in 2015 were not associated with any mental disorder [17]. The same result was also noted by the National Institute of Statistics [18] on suicide in the Italian population from 2011 to 2013, showing that more than 80% of cases are not associated with relevant mental disorders or physical illnesses. Hanging, self-poisoning, and the use of firearms are the most common methods of suicide [2,19]. Firearm possession can facilitate the transition between suicidal ideation and the enactment of suicidal behaviour [20,21].

Many studies have investigated suicide both in the general population and concerning specific professions such as help professions, including the police force [22,23,24,25,26]. Much of the scientific literature has shown a higher incidence of suicide rates among police forces than in the general population [22,27,28,29]. Research has focused on identifying potential risk factors for suicide, including an attempt to investigate the primary motivations for women and men in uniform to engage in suicidal behaviour.

A recent study investigated the perception of suicidal events by police personnel, showing an appropriate awareness of the suicide problem on behalf of Italian State Police workers [30]. Another study evaluated the suicide cases in the Italian State Police between 1995 and 2017, showing a trend of reduction in suicide cases in 1995–2007, followed by a stable trend until 2017 [22]. Another evidence of a steady trend was also underlined for 2015–2020 [23].

The period of intense and prolonged stress experienced for the COVID-19 pandemic has globally affected mental health [31,32,33]. The phenomenon involved the general population and helping professionals such as doctors, nurses, social and health workers, and police officers employed on the front line to deal with the ongoing health emergency [34,35,36,37,38]. During pandemics, the role of the police force proves to be crucial in policing public order and enforcing governmental measures aimed at containing the contagion. Recent studies have addressed the psychological impact of the SARS-CoV-2 pandemic on police officers. An analysis performed on security forces in Spain showed a significant level of burnout during the first wave, with high levels of emotional exhaustion, depersonalisation, and loss of personal development among the law-enforcement personnel [39]. A European multicentre study conducted during the first pandemic phase through an online survey showed a good level of stress tolerance among police officers [40]. In addition, recent research on a sizeable law-enforcement sample during the early phase of the pandemic in China showed a higher perception of job-related stress in women, higher health risks, psychological distress, and work-related stress in chronically ill workers, with a reduced health risk in younger personnel [41].

The pandemic is posing an occupational stressor for law enforcement personnel. Therefore, the need to quantify this phenomenon and put supportive programs in place is a high priority [42]. During the pandemic period, the Italian State Police implemented different support programs for the personnel. These included a national toll-free number to provide information on COVID-19 to police staff, a national health care service, vaccination services, and telematic working modalities. 

Furthermore, following the indications of the international scientific community on the psychology of emergencies, the Central Directorate of Health has launched the “Insieme Possiamo” (Together We Can) project, a psychological intervention protocol. This project aimed at developing, through an offer of services, both in presence and online, direct contact and active psychological support throughout the territory, offering the opportunity for advice and intervention by psychologists of the Italian State Police. A space for listening and psychological advice at a distance can be accessed online, through the platform “Doppiavela”, even anonymously with a nickname, and obtain answers to their personal, professional, or family needs.

The main hypothesis of this study is that during the pandemic period, the suicide rate among police personnel may not have increased.

Our study firstly aims to perform a descriptive analysis of the suicide in the Italian police from 2016 to 2021, and secondly aims to compare the pandemic and pre-pandemic periods.

## 2. Materials and Methods

### 2.1. Data Source

The Department of Public Safety of the Ministry of the Interior provided the annual number of active police personnel, stratified by age, gender, and ethnic group. Police personnel suicide deaths were analysed by reviewing files for the study period (2016 to 2021). There were no significant changes in the population of active police over the considered period.

The files were collected at the Centre of Neurology and Psychiatry, and data were extracted from 1 January 2016 to 31 October 2021. Each file contained the medical records of the subjects, from the time they first joined the State Police until the date of the suicide event, their marital status, level of education, territorial differences, the method used to commit suicide and the motivation for the suicidal act. Two independent reviewers analysed each file based on a standardised form used for the abstraction process. Data were collected at the time of hire and completed suicide for staff without mental disorders. A surveillance procedure with recurrent psychiatric visits was in place for those diagnosed with mental disorders.

This made it possible to assess the presence of any mental disorder diagnosed during police work and for which the employee may have been declared temporarily unfit or under medical supervision. Where no information was collected, the information on psychopathology aspects was taken from the reports of the competent State Police doctors. After each case of suicide, an intervention of psychological support to family members has been activated by psychologists of the Italian State Police. In this context, further information on family conflicts or emotional problems was collected immediately after the suicide.

This study was conducted according to the guidelines of the Declaration of Helsinki. The survey was approved by the competent Ethics Committee (Prot. #0036646).

### 2.2. Statistical Analyses

We used the SPSS Statistics V27.0 software (IBM Corporation, Armonk, NY, USA, 2021). We have considered the marital status, level of education, territorial differences, the method used to commit suicide and the motivation for the suicidal act. The suicide rates for police personnel were calculated as the number of suicides per half-year per 100,000 active police personnel units. To summarise and analyse trends, we performed a regression using the Joinpoint Regression Program, version 4.9 (https://surveillance.cancer.gov/joinpoint/ (accessed on 25 March 2021)). We analysed the average half-year percentage changes during the entire period (2016–2021). We used the half-year suicide rate as the dependent variable, assuming constant variance (homoscedasticity) and logarithmic transformation. We set a maximum number of 2 joinpoints and used a permutation test with an overall significance level set to *p* < 0.05.

## 3. Results

From 1 January 2016 to 31 October 2021, there were 65 cases of death by suicide among police personnel, of whom 62 were men and 3 were women, corresponding to 0.066% of the average population of Italian State Police personnel during the study period (2016–2021). The mean age of people dying from suicide was 47.54 years (SD = 8.205, range = 23–60). 

Regarding marital status, at the time of death, 56.9% of the sample were married (37), single in 20% of cases, separated/divorced in 16.9% (11), and 4.6% (3) cohabiting. 

Concerning the level of education, 66.2% of cases (43) had high school graduation, 16.9% had a middle school certificate (11), and 7.7% had a college degree (5). In comparison, in 9.2% of cases (6), the level of schooling was unknown or not indicated.

With regard to the territorial differences, the northern regions report a higher percentage of suicides (44.6%), followed by the regions of the South and the Islands (33.8%). In comparison, lower values are reported in the areas of Central Italy (21.5%).

The average police suicide rate during the period 2016–2021 was 10.7 per 100,000 individuals per year (95% CI = 9.54–11.86). The average suicide rate between 2016 and 2019 was 12.10 per 100,000 individuals per year (95% CI = 10.86–13.34), and the average suicide rate between 1 January 2020 and 31 October 2021 was 7.88 per 100,000 individuals per year (95% CI = 4.97–10.79). 

Police personnel committed suicide by using their service weapon in 83.1% of cases, hanging (6.2%), defenestration/precipitation (4.6%), using their own gun (3.1%), and a stabbing weapon (3.1%). Overall, firearms were used in 56 cases (86.2%). 

Our data showed the reasons behind the suicide during the period under study, which consisted mainly of emotional problems (38.5%) (arising from personal and family issues, grief, separation/divorce), psychiatric disorders (15.4%), organic diseases (6.2%), economic problems (3.1%), perceived dishonour (1.5%), and unknown motivations (35.4%). In particular, the analysis of the pre-pandemic period (2016–2019) showed that the reasons were mainly emotional problems (40.8%) (arising from personal and family issues, grief, separation/divorce) and psychiatric disorders (16.3%), while in 32.7% of cases they were unknown. Additionally, regarding the pandemic period, the motives were predominantly related to emotional problems (31.3%) and connected to psychological (12.5%) and organic diseases (12.5%). However, the number of cases in which it was impossible to ascertain the motivation for suicide was still high (unknown, 43.8%).

Joinpoint regression analyses for 2016–2021 showed that suicide cases in the Italian police personnel showed a not significant decrease, with an average half-year per cent change of −3.9 (t = −0.8; Prob > |t| = 0.424).

A model with one Joinpoint showed a period from January 2016 to December 2019 with an average not significant half-year percent increase of 12.3 (t = 1,4; Prob > |t| = 0.209), followed by a period from December 2019 to October 2021 with an average not significant half-year percent decrease of 31.6 (t = −1.9; Prob > |t| = 0.098) (Table 1 and Figure 1).

## 4. Discussion

World Health Organization report up-dated to 19 November 2021, listed 255,324,963 cases of SARS-CoV-2 worldwide since the beginning of the pandemic, including 5,127,696 deaths [34]. As some reports point out, from the very first months of the health emergency, there was a rapid and significant increase in the prevalence of mental health problems both nationally and globally compared to the condition before the pandemic outbreak, mainly including stress-related disorders, depression, anxiety, and sleep disturbance [43,44,45,46]. Other psychological and psychiatric conditions related to the pandemic emergency were loneliness, alcohol use problems, somatic symptoms, panic, adjustment disorders, suicidal ideatioobsessive-compulsive symptoms, phobic anxiety, and suicidal ideation and attempts [35]. Mental health problems mainly affected doctors, nurses, social and health workers, and police officers [34,35,36,37,38]. The need to monitor the possible mental health consequences of the pandemic was also highlighted in suicide deaths [47,48]. We found a higher rate of suicide in northern Italy. We found a higher police rate of suicide in northern Italy (44.6%) during the period considered in this study. Considering that the police population is homogeneous throughout the country, this may reflect the general population trend, which is also higher in north Italy [17].

Suicide is a significant public health problem investigated in the general population and specific occupations such as the helping professions [22,23,24,25,26]. It has been found that psychiatric disorders, suicidal ideation, and suicide are considerably higher in doctors than in the general population [49,50,51]. Other studies have shown higher suicide rates among police officers [22,23,27,28,29]. A steady trend in suicide rates in the Italian State Police was found between 2015 and 2020 [23].

Italy was among the first countries to be affected by the COVID-19 pandemic. Since the first measures that determined the lockdown of the country, law enforcement personnel have been employed in the activities necessary to cope with the ongoing emergency, with a consequent increase in possible stress factors arising from the management of social restrictions imposed by governments. 

Given the specific activities carried out by police officers in this context, we wanted to explore suicide rates among police officers in the pre-pandemic and pandemic periods and the characteristics of suicide events.

As an initial result, this work has shown that during the pandemic (2020–2021), the suicide rate in the State Police was lower than that of the pre-pandemic period (2016–2019). The trend of the suicide rate for the two periods considered in this study (i.e., the pre-pandemic period 2016–2019 and the pandemic period 2020–2021) showed a not significant decrease during the pandemic period. Other studies of different populations have shown that suicide rates changed in the pandemic era [52,53,54]. Although suicide is an unpredictable event, there are various signs, including psychological and behavioural changes and psychiatric symptoms, that can be used to assess risk. It should be noted that effective prevention strategies can also be implemented through awareness and support programs [55,56], virtual health visits support, new technologies [57], and on anonymous bases [58,59].

Suicide prevention is a priority, especially now, as collective quarantine measures have been linked to an increased risk of suicide, and quarantine has been associated with adverse effects on mental health [60,61]. 

The COVID-19 pandemic has profoundly altered the rhythms and styles of emotional, work and social coexistence of millions of individuals following the measures adopted by the Italian government to deal with the health emergency. During quarantine periods, forced cohabitation and limited social interactions can worsen family conflicts, and social isolation can increase fear and insecurity [62]. 

Some studies have investigated emotional cohabitation and changes in cohabitation following catastrophic events, including the effects of different living styles during the lockdown in Italy on dimensions such as stress and coping strategies. Shared coping strategies (dyadic and collaborative) reflect stress management as a collective and not an individualistic process. Each individual does not manage stress in a vacuum but within an interpersonal context [62,63].

Given the possible consequences of the health emergency, health professionals have implemented awareness-raising campaigns for suicide prevention, focusing on understanding emotional crises [61]. In this context, implementing staff support services may have positively affected the suicide rates.

During the pandemic period, the Central Directorate of Health of the Italian State Police has promoted enhanced psychological counselling and support programs for police personnel to cope with the burden of stress due to the pandemic period, uncertainties and worries for their loved ones. The measures implemented aim to provide adequate support and facilitate access possibilities to mental health professionals, overcoming the difficulties related to the stigma that can lead to delayed access to care. Recourse to a mental health specialist can be seen as a sign of weakness for those who are experiencing psychological distress, even related to personal/family situations. In addition, police personnel might fear that a psychiatric evaluation might negatively affect their work, including restrictive measures (e.g., disbarment), reassignment to other duties, lack of promotion, and stigmatisation.

The possibility of access to psychological support programs for law enforcement officers and technology can be considered a sign of cultural change towards the stigma towards psychiatric pathologies. 

Other aspects could have limited the suicide rates in the Italian State Police, including a sense of altruism to keep working to help the community, a mind of usefulness to the national assembly, good staff resilience, and other factors that deserve further investigation.

Regarding risk factors for mental disorders in police personnel, particular attention should be focused on stress factors [64]. Many studies have shown a high risk of stress for workers employed in the police force [65,66,67,68]. Major stressors could be related to specific operational tasks (e.g., patrol activities, traffic control, criminal investigations, crime prevention, community service) and organisational tasks (e.g., personnel/human resource selection and management, recruit training, public information, bureaucratic procedures) [69]. Specifically, operational stressors relate to the characteristics inherent in the type of work performed, which involves using force, decision making in critical situations, the risk to one’s safety and colleagues, risk of accidents, exposure to suffering, and scenes of violence [70]. There is evidence of different factors that primarily affect the psychological distress of police officers. Some studies highlighted the role of operational aspects (for example, patrol tasks, traffic control, criminal investigations, crime prevention, and community services) [71,72,73,74], others a more significant influence of organisational stress factors (for example, selection and management of personnel/human resources, training of recruits, public information, bureaucratic procedures) [68]. Other clinical and personal aspects, including a diagnosis of a mental disorder, personality traits, and critical events related to the family sphere may influence the psychological sphere of police officers who commit suicide [75,76,77,78,79]. 

This study showed that disturbance in interpersonal/affective relationships constituted the central theme that suicidal patients identify as making life meaningful. These aspects are among the main points for suicide prevention programs, including community mental health services. Further research should be focused on the role of stress and other risk factors for suicide in the law-enforcement during the pandemic.

*Limitations*. The main limitations of this study consist of its descriptive and retrospective nature, so the results should be taken with appropriate caution. A second limitation is that, to date, there is a lack of studies on the suicide trends during the pandemic in the law-enforcement and in other similar groups of workers. Furthermore, our findings are not fully comparable with countries without the same resources implemented as Italy.

## 5. Conclusions

In conclusion, this study showed that the suicide rate among the Italian Police personnel during the pandemic period did not increase. Over the 2016–2021 period, the overall suicide rate for police officers had a stable trend. When comparing the pre-pandemic period to the post-pandemic period, a not significant trend of increase was found in the 2016–2019 period and a not significant decrease in the 2020–2021 period. Difficulties in interpersonal/affective relationships constituted the central theme that suicidal patients identified as making life meaningful. The implementation of staff support services may have positively affected the phenomenon. Other interpersonal, social, psychological, and clinical aspects may have influenced the trend, which is a point that deserves further investigation.

## Figures and Tables

**Figure 1 ijerph-19-05904-f001:**
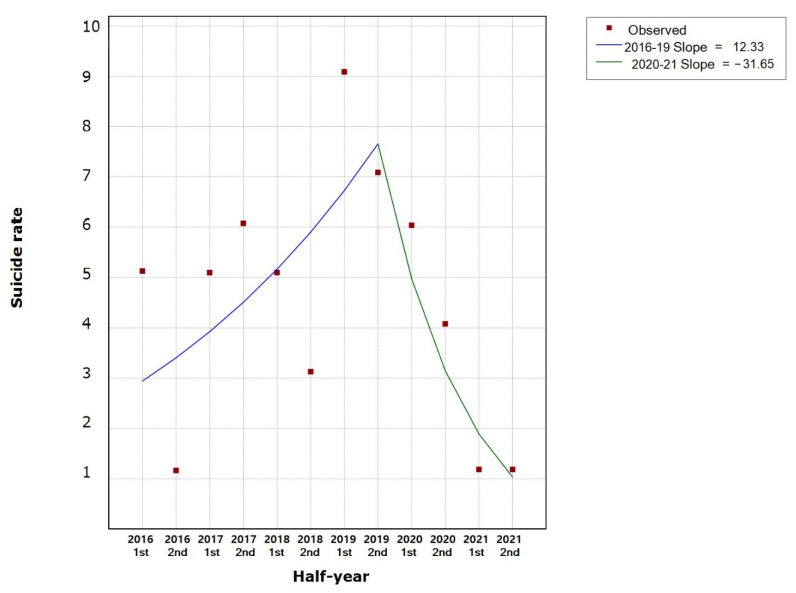
The trend of suicide rate (per 100,000 population) during the study period.

**Table 1 ijerph-19-05904-t001:** Suicide trends during the study period with half-year per cent change (HPC).

Segment	Lower Endpoint	Upper Endpoint	HPC	Lower CI	Upper CI	Test Statistic (t)	Prob > |t|
2016–2021	January 2016	October 2021	−3.9 %	−13.6	6.9	−0.8	0.424
2016–2019	January 2016	December 2019	12.3 %	−7.9	37.0	1.4	0.209
2020–2021	December 2019	October 2021	−31.6 %	−57.3	9.4	−1.9	0.098

## Data Availability

Data from this research are kept in the archives of the Central Directorate of Health, Central Operational Health Service—Centre for Neurology and Psychiatry of the Department of Public Safety—Ministry of the Interior.

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
