# Peer review of "Suicide Trends in the Italian State Police during the SARS-CoV-2 Pandemic: A Comparison with the Pre-Pandemic Period"

_ijerph, 2022, doi:10.3390/ijerph19105904_

Round 1

Reviewer 1 Report

Dear Authors,
The topic is interesting. The literature review was well conducted. The research process is not objectionable. The inference is correct. I recommend the article for publication.

Reviewer 2 Report

The manuscript with the title "Suicide trends in the Italian State Police during the Sars-Cov-2 2 pandemic: A comparison with the pre-pandemic period" is very well presented. The background is comprehensive but I have some minor suggestions for the discussion and conclusion part:

1-Page 3 line 145: 

"From January 1, 2016, to October 31, 2021, there were 65 cases of death by suicide among police personnel, of whom 62 were men and three women. " could you please provide the total population here or the percentage that will give a better view of the situation. 

2- on page 5 line 202: 

"We found a higher rate of suicide in northern Italy." this statement is not clear, does it mean in the general population or just in the police department? Also could you please clarify when you have noticed this trend? during the pandemic or in general.

3- page 6 line 219: 

"As an initial result, this work has shown that during the pandemic (2020-2021), the suicide rate in the State Police was lower than that of the pre-pandemic period (2016-2019). " When I read this statement I expected to see another comparison results. but there is not such a statement following the results. then in conclusion you began the paragraph with the claim that the pandemic didn't affect the suicidal rate among police officers: 

"this study showed that the suicide rate among the Italian Police personnel during the pandemic period did not increase." 

could you please clarify this? 

4- To conclude the study it will be interesting to discuss your opinion if the stress didn't affect the police officer's suicidal rate, what can be the factor? does it mean that the government handled the mental health of police officers perfectly?

The presented data will become more valuable if you can compare them with maybe other similar studies in the other countries regarding the suicidal rate of police officers during the pandemic. or maybe you can compare these results with a similar situation like the suicidal rate (either increase or decrease) among the health care professionals.

Also, the data that have been presented didn't specify the department where the officers were working. It can be a difference in the suicidal rate among clerical police officers and those who work in the field. 

Author Response

This manuscript is a resubmission of an earlier submission. The following is a list of the peer review reports and author responses from that submission.

Round 1

Reviewer 1 Report

Dear Authors,

The topic is extremely interesting. The subject matter of the article is very important. Unfortunately, the research problem has not been treated with scientific maturity. The research methodology is at a very basic level. The conclusions are also laconic. The article is unfortunately not suitable for publication.

Reviewer 2 Report

This study aimed to describe suicide events that occurred among the Italian State Police over a 5-year period and to compare the suicide rates pre- and during-Covid. Using data from available records provided by the Department of Public Safety, details regarding each suicide event were collected and summarized.  More specifically, basic demographic data, geographic information, methods used to commit suicide and potential motivations were collected for a 5-year period.  The authors reported that suicide rates declined during COVID and suggest that the decline may be due to programs implemented by the Central Directorate of Health of the Italian State Police. 

Overall, I applaud the authors for investigating rates of suicides among a group of essential workers at high-risk.  The authors also provide a nice review of the literature on this topic in their introduction; however, the main weakness of this research is it's descriptive nature and crude analyses of data.  Suggestions for improvement are described below.   

Abstract

Lines 19-21 of the abstract are somewhat confusing.  The abstract highlights services related to covid education and vaccination as being the main support program instituted by the Italian State Police.  However, it is unclear as to how much, it at all, this would impact suicide rates.  It seems that the project aimed at providing psychological services as described in line 94 would have a greater impact, but there is no mention of this specific program in the abstract.

Line 27 – Please clarify what is meant by “individual psychological factors in response to the pandemic may have positively affected the phenomenon”.

Methods

Line 107 – It would be helpful for the reader to know how large the police force is and if there were any changes in the population of active police over the time-period of interest.  If characteristics of the police population changed pre and during the pandemic, perhaps this could help explain the findings.    

Line 111- Please provide more detail about the data abstraction process. Specifically what data elements were abstracted and how was the data abstraction done (i.e., was each file reviewed by a single individual or were there multiple reviewers?  Was there a standardized form used for the abstraction process?)    Regarding the data elements- I do see specific elements listed in lines 124-125 but it is unclear if other elements (such as length of time in the force and assigned roles) were collected but not analyzed.

Line 112- The authors state “Each file contains the psychodiagnostic anamnesis of each operator, from the time he/she first joined the State Police until the date of the suicide event.”  Please provide more details.   Is this something that is collected yearly on each member of the police force or only when hired and then when a concern is raised?  Also, did you mean ‘officer’ instead of ‘operator’?

Line 118- Suggest clarifying the sentence “Further information on family conflicts or emotional problems was collected immediately after the suicide.”  How was this done and by who?  Also, the current wording suggests that this was actually done as part of the study, rather than abstracting the data from existing reports.

Lines 123-127- The authors performed basic descriptive analyses, but were more robust analyses methods considered?  If the authors have not already done so, I would recommend consulting with a statistician to determine if more sophisticated analyses could have been performed to help support their conclusions.  

Results

Lines 129-139 –  Were these factors similar during the two time periods (pre and pandemic)?

Line 137-138 – The regional differences are interesting but is this because northern Italy has a larger population of officers? If not, do the authors have any idea what might explain these differences?  Were these regional differences similar pre and during the pandemic?  

Line140-143 – Consider adding a graph to show suicide rates over time, perhaps by quarter or six-month periods

Line 150-  The method section should mention that files were obtained from the Centre of Neurology and Psychiatry.

Line 157 – Were the differences pre and pandemic periods significant?

Discussion Section

Line 164 – The sentence should specify these are the number of COVID cases and deaths worldwide.

Line 232-233 – This sentence seems to overstate what was done and should be reworded to align more closely with the stated aim of the study which was descriptive in nature.

Line 262-  It is not clear how access to an anonymous service may have “produced attitudes of greater trust in the professionals of the police health service who gave them support”.  As this is not discussed or mentioned anywhere else in the manuscript, I recommend removing this part of the final sentence. 

A paragraph discussing study limitations should be added. 

Other potential reasons for the decline in suicide rates should be considered in the discussion section. 

It is unclear if the project aimed at providing psychological services started during COVID or existed pre-COVID but was promoted more during COVID. If it existed pre-COVID it would be helpful to know if utilization of this program increased between the two periods - if it did this would help support your conclusion that the support program may have resulted in the decline in suicides. 

Reviewer 3 Report

The authors aim to describe the psychological impact of the COVID-19 pandemic on the suicide rates within the Italian police force, comparing rates pre- and post-pandemic. This is an interesting topic and much-needed area that needs further attention and research on. However, the manuscript in its present state does not make much sense/reads disconnected, and the data provided is very limited and weak. Offered below are both general and specific comments:

Abstract: “smart working modality” What is meant by this? Is it telecommuting or working remotely?

Abstract: Would just call it suicide rather than “suicide phenomenon”; no need for the phenomenon.

Lines 32-37: When considering deaths outside from disease, right? The way it is stated now makes it seem like suicide is above cardiovascular disease and infectious diseases.

Line 64: Does events include attempts as well? Unclear if it’s counting both suicides and suicide attempts or just the former.

Line 124: Are the characteristics being used as control variables? Unclear.

Results: Demographics would be better presented as a table

Results: Periods instead of commas for numbers. 7.88 not 7,88

Line 146: What is the precipitation referring to?

Lines 149- 162: Confused about whether these statistics come specifically from the same cohort that was analyzed for suicide or just looking at the population generally?

Discussion/Lines 183-189: Are there any other potential confounding or contributing factors to this? Can the authors truly be sure that the reduction in suicides was due to the psychological resources given to them at work? What if they had internal pressure or a sense of altruism to keep working to help the community because they felt too many people were dying so they could not afford to kill themselves, even if they had suicidal ideation? Not wanting to let others down. What about culture? I am not convinced by the descriptive statistics and that the call line alone managed to reduce suicide rates.

Line 194: “Suicide is un unpredictable event” It can be but there are signs, including depression and changes of behavior that can indicate this

Line 196: Is telematics the same as telehealth or virtual health visits?

Line 247: “There is inconsistent evidence of the factors…” not mutually exclusive so would not say there is inconsistent evidence, but both could be true (operational factors and organizational factors). The relationship between the two is still not fully understood.

Lines 26-258: Did not really get this sense from the entire paper since it mostly focused on the community health services. The discussion tries to talk about the social support that potentially drove down suicide rates but that was not clear to me in the results and feels really disconnected from the data presented. The argument needs to be strengthened and with further evidence.

The data is very sparse and lacking. It would be good to do additional analysis and even do some chi-square or other appropriate statistical tests to see if the implementation of these community hotlines were statistically significant. After reading the whole thing it feels like two arguments are being made: 1) support services helped keep suicide rates down. The results attempt to support this but not really, 2) individual psychological factors made suicide less likely but the data very weakly/does not support this at all.

No quotes needed around the sections on Author Contributions, Funding, Conflicts of Interest.

Round 2

Reviewer 2 Report

While the authors responded to concerns that were raised, there are still opportunities for improvement.  

Unfortunately, I found the additional analyses and the tables and figure that were added to be confusing. For example, it is unclear what is being shown on the x and y axes in the graph.  Better labeling and explanation in the text may help the reader better understand what is being presented.

Reviewer 3 Report

It is appreciated the time and effort taken by the authorship to revise the manuscript. I found it to be significantly improved. The introduction is much clearer and more cohesive now. The methods and results are more robust. I only have a few more minor comments:

-Commas used in numbers where decimals should be and other times decimals used where commas should be. Will let copy editing decide though.

-Table 1: Are the numbers in the columns percentages? Please be more clear in the labeling by either modifying the title of the table or indicating in the first row. 

-No quotes needed around the Study approval statement from the ethics committee. 

-Additional limitations is that for countries without the same resources implemented as Italy, the findings on suicide rate may not be generalizable. 

-